# A method to build extended sequence context models of point mutations and indels

Jörn Bethune[1,2,4], April Kleppe ®[1,2,4] & Søren Besenbacher ®[1,2,3] ✉

The mutation rate of a specific position in the human genome depends on the sequence context surrounding it. Modeling the mutation rate by estimating a rate for each possible k-mer, however, only works for small values of k since the data becomes too sparse for larger values of k. Here we propose a new method that solves this problem by grouping similar k-mers. We refer to the method as k-mer pattern partition and have implemented it in a software package called kmerPaPa. We use a large set of human de novo mutations to show that this new method leads to improved prediction of mutation rates and makes it possible to create models using wider sequence contexts than previous studies. As the first method of its kind, it does not only predict rates for point mutations but also insertions and deletions. We have additionally created a software package called Genovo that, given a k-mer pattern partition model, predicts the expected number of synonymous, missense, and other functional mutation types for each gene. Using this software, we show that the created mutation rate models increase the statistical power to detect genes containing disease-causing variants and to identify genes under strong selective constraint.

The germline mutation process is the source of all genetic variation, including all adaptive and deleterious variants. Understanding and modeling this process can be used to calibrate variant calling[1], infer demographic history[2], infer patterns of genome evolution[3], identify sequences of clinical relevance for human diseases[4], and infer selective constraints of genes[5]. The main factor determining the mutation rate of a given site in the human genome is the sequence context surrounding the position. For example, spontaneous deamination of methylated cytosines results in ten times higher C-to-T rates at CpG sites[6–9]. Other factors that affect the mutation rate—such as GC content, CpG islands, epigenetic modifications—are associated with varied mutation rates depending on nucleotide context[10] and thus cannot be studied without taking sequence context into account.

Since nucleotide context is the main determinant of the mutation rate it is relevant to make models that estimate the mutation rate of a position using that feature. Previous studies estimated such rates by assigning an independent rate to each k-mer[10,11] or using a logistic regression model with a dummy variable for each nucleotide at each k-mer position[12] and allowing interactions between at most four positions. Both these strategies have been used to build models that predict mutation rates using 7-mer contexts but become infeasible for longer contexts. Furthermore, these previous studies focused on point mutations, and little effort has been given to estimating the position-specific rate of germline indels. Neither Carlson et al.[10] nor Aggarwala and Voight[12] consider indels. Samocha et al. calculate the rate of frameshift indels per gene but do not model the indel mutation probabilities at each site[11]. Instead, they estimate the rate of frameshift indels in a given gene by assuming that the rate of such variants is proportional to the rate of nonsense mutations. While some correlation exists between the number of polymorphic nonsense mutations and the number of polymorphic frameshift indels ($r = 0.493$, see methods) there is no observable correlation between the number of de novo nonsense mutations and de novo frameshift indels ($r = -0.003$, see methods). This lack of correlation among de novo variants

[1]Department of Molecular Medicine (MOMA), Aarhus University Hospital, Aarhus, Denmark. [2]Department of Clinical Medicine, Aarhus University, Aarhus, Denmark. [3]Bioinformatics Research Centre, Aarhus University, Aarhus, Denmark. [4]These authors contributed equally: Jörn Bethune, April Kleppe. ✉e-mail: besenbacher@clin.au.dk

indicates that the correlation between nonsense mutations and frameshift indels in segregating variants is primarily due to selection and not mutation. This makes the effectiveness of the strategy of calculating the rate of frameshift indels based on the rate of nonsense mutations doubtful. Since frameshift indels account for 44% of the LoF mutations in human genes[5], making models that can predict the rates of such mutations should be considered an important task. In this work, we present a new method that can predict the mutation rate of both point mutations and indels based on nucleotide context.

## Results

To overcome the problem that many k-mers will have few or zero observations if we use long k-mers to predict the mutation rates, we propose a method that we call *k-mer Pattern Partition* (kmerPaPa). The main idea is that we partition the set of all k-mers by a set of IUPAC patterns so that each k-mer is matched by one and only one of the IUPAC patterns in the set. This partition should be done so that a pattern matches k-mers with similar mutation probabilities. Figure 1 shows how the 16 possible 3-mers containing C→G mutations can be partitioned using 10 patterns. An exact algorithm to calculate the pattern partition that optimizes the loss function is presented in the methods section. The loss function contains two regularizing hyperparameters $c$ (complexity penalty), and $\alpha$ (pseudo count), which are fitted using twofold cross-validation.

### Testing prediction of point mutation probabilities

To make models that only reflect the mutation rate and have as little bias from selection and biased gene conversion as possible, we use observed de novo mutations as training input to our model (see methods). To test the model, we first separate the data into a test and training set, using the even-numbered chromosomes for training and the odd-numbered as test data (see Supplementary Fig. 1 for an overview of the data used in different analyses). We then fit independent pattern partition models to each of the six mutation types (A→C, A→G, A→T, C→A, C→G, C→T) for different values of k. We compared the performance of kmerPaPa to a model that assigns a rate to each k-mer (called "all k-mers" in Fig. 2). As with kmerPaPa we also include a pseudo count in the "all k-mers" model and fit its value using 2-fold cross-validation. Figure 2a shows the test out-of-sample performance

of these models for different values of k. For kmerPaPa, the joint Nagelkerke $r^2$ across the 6 mutation types keeps increasing as k increases. Whereas the alternative "all k-mers" model begins overfitting at 5-mers and performs poorly for larger values of k. The results reveal that nucleotides situated more than three base pairs away can affect a position's mutation rate as the 9-mer partition outperforms the 7-mer partition in four of the six different mutation types, even though the 9-mer models tend to include fewer patterns (see Supplementary Fig. 2).

### Testing prediction of indel mutation probabilities

Besides point mutations, it is essential to consider insertions and deletions. We assign mutation probabilities to indels by looking at the k-mers around the breakpoints - one for insertions and two for deletion (start and end). However, unlike point mutations, it is often impossible to precisely determine the position of an indel. Usually, more than one possible mutation event could have changed the reference sequence to the alternative sequence. If, for instance, "CAG" is changed to "CAAG," it is impossible to know whether the DNA break and base insertion happened between C and A or between A and G. We handle this uncertainty by enumerating all the possible positions and randomly selecting one of them for each observed event (Fig. 3). We evaluate the performance of the predicted indel rates using the same test/train split as we used for point mutations (Fig. 2b). Both for all-k-mers and kmerPaPa the out-of-sample performance increases with increased k. But the kmerPaPa model outperforms the all-k-mers model for all values of k.

After validating that the extended k-mer models built by kmerPaPa give good results on an independent test set, we trained new kmerPaPa models on the whole data set. The following sections use these models to investigate the sequence contexts that increase mutation rates, find genes harboring disease-causing variants, and quantify the intolerance to mutations of human genes.

### Identification of patterns with unusually high or low mutation rates

An advantage of a k-mer pattern partition compared to a regression model is the direct interpretability of the output. By ordering the output patterns based on their mutation rate, we can directly observe

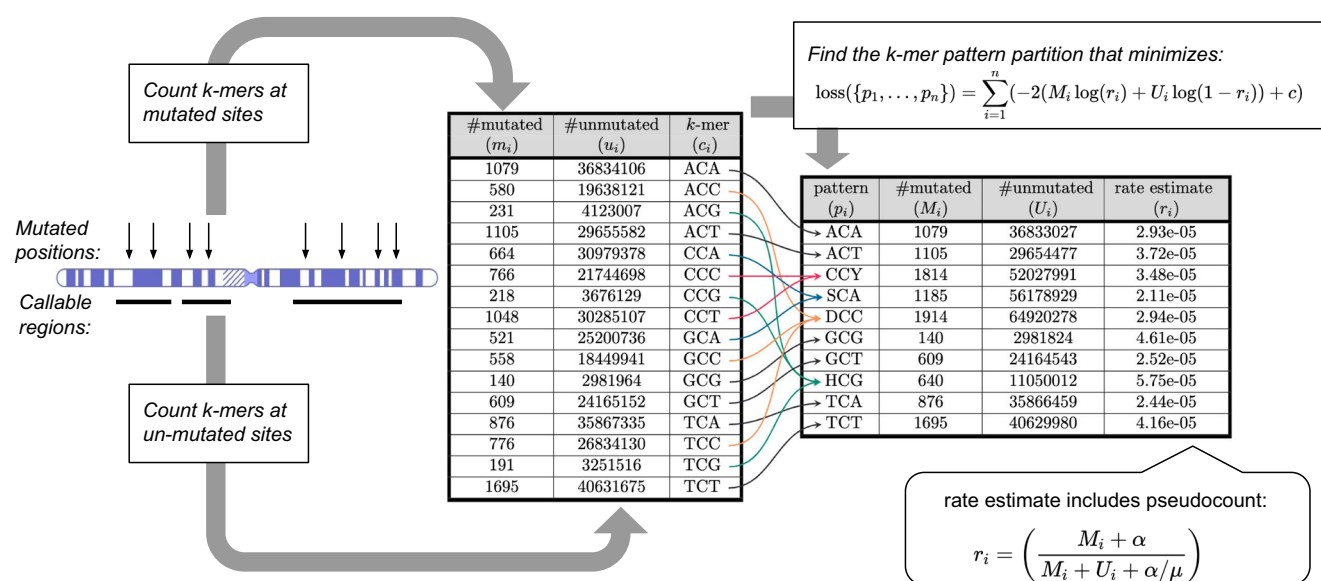

**Fig. 1 | Overview of k-mer pattern partitioning (kmerPaPa) model.** Input data consists of a list of observed mutations (here C→G mutations) and a bed file with regions sufficiently covered by WGS data to detect mutations. From this, we calculate a table with the number of times each possible k-mer is observed with the central base mutated and unmutated. The k-mers are then grouped using a set of IUPAC patterns so that each k-mer is matched by one and only one pattern. Out of the exponentially many possible pattern partitions the one that minimizes the loss function is chosen.

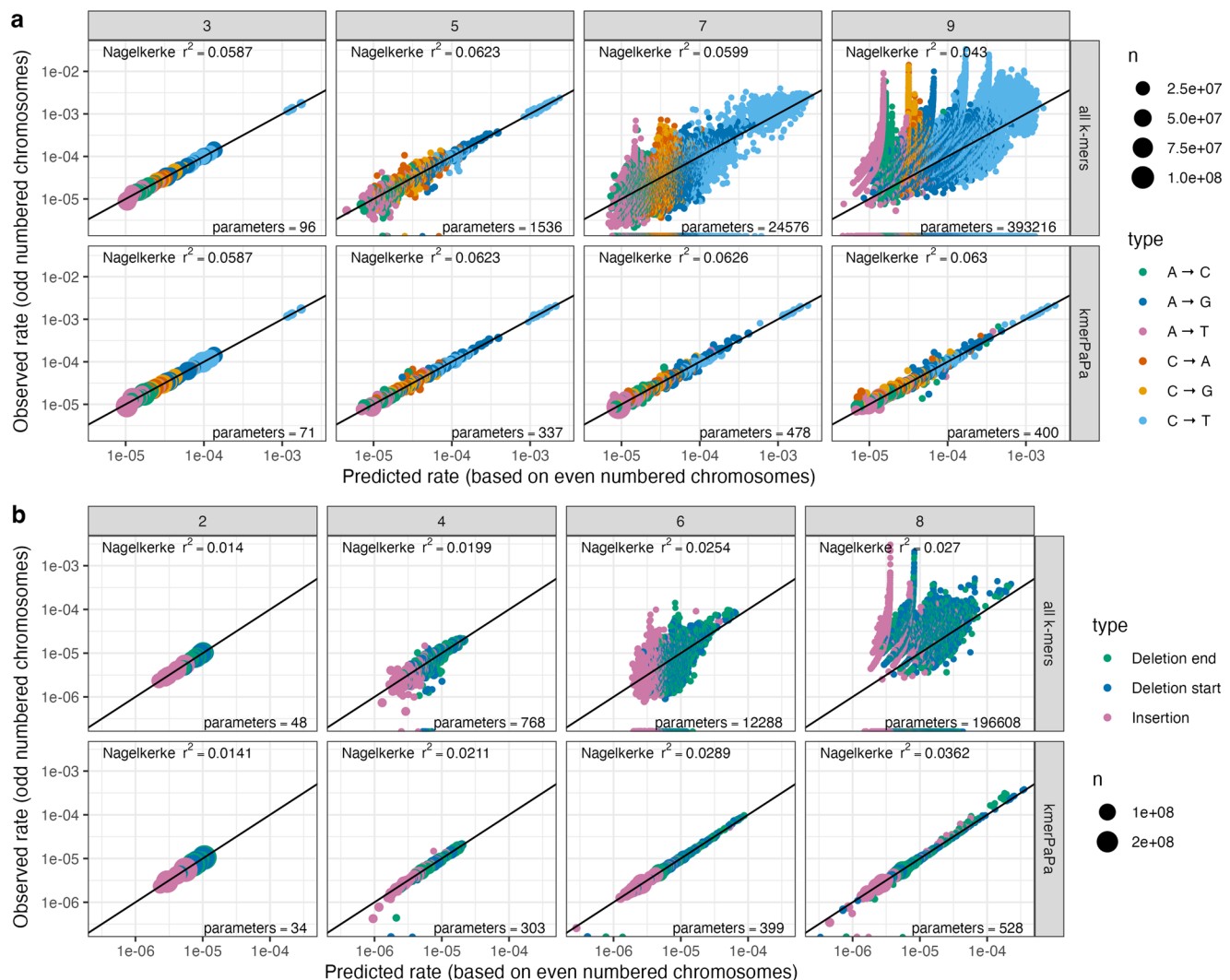

**Fig. 2 | kmerPaPa performance on test data compared to predicting a rate per k-mer.** All plots show predicted rates on training data (even-numbered chromosomes) on the *x* axis and observed rates on test data (odd-numbered chromosomes) on *y* axis. The first row ("all k-mers") shows the rates for each k-mer. The second row ("kmerPaPa") shows the rate for each k-mer pattern. The sizes of the points reflect the number of times the k-mer/pattern is observed in the genome. **a** The predictions for point mutations for four different values of k. **b** The predictions for indels for four different values of k.

patterns with extreme mutation rates. Figure 4 shows the relative mutation rate of each of the patterns for each of the mutation types (blue points) and the rate of each 3-mer (red points). For most mutation types the range of the predicted mutation types using k-mer patterns is an order of magnitude larger than those predicted by 3-mers. Many of the patterns with high point mutation probabilities match the patterns previously reported by Carlson et al.[10] and Aggarwala and Voight[12]. For instance, the top pattern for A→G mutations "YYCAATG" match the CCAAT motif reported by Carlson et al.[10] and the YCAATB pattern reported by Aggarwala and Voight[12].

Previous studies have reported that polymerase slippage at short tandem repeats is responsible for 75% of indels[13]. It is thus not a surprise that many of the indel patterns we observe contain repeated sequences. For insertions, we observe the highest mutation rate for (T) n and (A)n mononucleotide repeats followed by (G)n and (C)n repeats. For the deletions, the top patterns correspond to (AG)n and (CT)n dinucleotide repeats. After these, we observe a high deletion rate of the middle two (underlined) bases in the palindromic sequence: CACATGTG. This sequence has not previously been described as a mutation hotspot and the reason behind its high deletion rate is unknown. One possible explanation is that the palindromic sequence can form a hairpin structure. DNA hairpins are known to cause

transient polymerase stalling, which leads to indel formation, whereas other alternative DNA structures (e.g., G4) cause persistent polymerase stalling and result in point mutations[14].

### Detection of genes where de novo mutations cause disease

Genes in which germline mutations cause disease can be found by looking for genes with surprisingly many de novo mutations in afflicted children. The k-mer pattern partitions we have created can be used to calculate how many mutations of a specific functional category (synonymous, missense, etc.) to expect in a given gene.

We have created a software tool—Genovo—to enumerate all the possible variants of a specific functional category and look up their mutation rates. Given a list of observed mutations, Genovo can then calculate p-values by sampling from the Poisson-Binomial distribution given by this list of rates (see methods and Supplementary Fig. 3). This functionality is similar to that provided by the tool denovolyzeR[15], so we compare our results to this tool.

As a first test, we compare the genic mutation rate predicted by Genovo and denovolyzeR to the number of segregating variants in each gene. First, we look at the number of rare (MAF < 1%) synonymous variants for each gene in the gnomAD database, where we observe a better correlation with the synonymous rate per gene estimated by

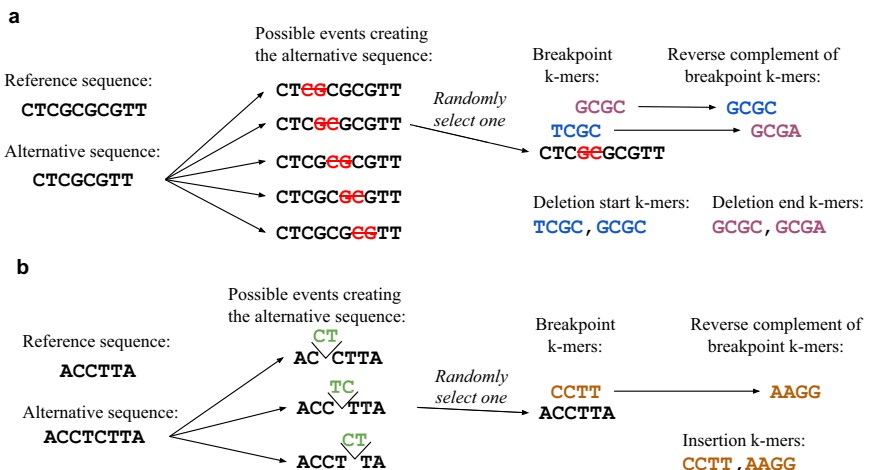

**Fig. 3 | Counting k-mers for indels.** The exact positions of indels are often inde-terminable based on the sequence data. To handle this we enumerate all possible events creating the alternative sequence and select one of them at random. We do this independently for each indel mutation. Unlike point mutations, we count both the observed k-mer and its reverse complement. **a** Deletion example. For deletions where we consider both the k-mer around the start breakpoint and the end breakpoint. The reverse complement of a deletion start k-mer is a deletion end k-mer and vice versa. **b** Insertion example.

Genovo ($r = 0.977$) than by denovolyzeR ($r = 0.958$) (Fig. 5a). Most of the variation in the number of synonymous variants per gene can be explained purely by the genes' length. But if we look at the rate per site and divide both the observed number and the predicted rate by the length of the coding sequence, we still observe high correlations ($r = 0.699$ for Genovo and $r = 0.679$ for denovolyzeR) (Fig. 5b). For nonsynonymous variants, we expect smaller correlations since differences in selective pressure between genes greatly influence these. For all classes, we observe better correlations with the rates predicted by Genovo than those predicted by denovolyzeR.

Secondly, we want to compare the statistical power of Genovo and denovolyzeR when it comes to identifying disease-causing genes. To do this, we look at 4293 trios from the Deciphering Developmental Disorders consortium[16] and divide the set of trios into an equally sized test and train data set. 350 genes contain at least one loss-of-function(LoF) mutation in the test data set. 45 of these show significant LoF enrichment after Bonferroni correction according to Genovo, and 33 of these are significant in the test data set. Running denovolyzeR yields 34 significant genes in the train set, 24 of which are significant in the test set. This means that Genovo has a higher validation rate in the independent test data set (73.3% compared to 70.6%) even though it identified a larger number of significant genes in the training data set.

### Quantifying genic tolerance to loss-of-function mutations

Besides finding genes where de novo mutations cause disease, we can also use our mutation rate models and the Genovo software to estimate the strength of negative selection acting on a given gene. To do this, we look for genes containing fewer segregating functional variants than expected because selection has purged deleterious variants from the population. The types of variants most likely to be deleterious are those that we expect to completely inactivate a gene, such as stop-gain, essential splice, and frameshift variants. The observed number of such loss-of-function (LoF) variants for a gene divided by the expected number predicted by our mutation rate model—the LoF O/E ratio—provides an estimate of evolutionary constraint. If a gene has an LoF O/E ratio around one it indicates that it is evolving neutrally and that no substantial fitness cost is associated with losing the gene. In contrast, haploinsufficient genes where two functional alleles are essential for survival will have LoF O/E ratios at or close to zero. We have calculated this ratio for each gene using the observed mutations from gnomADv2

and compared them to the ratios reported for each gene by gnomAD. Looking at genes predicted to be haploinsufficient by ClinGen[17] we observe that 55.5 percent of the genes have a Genovo O/E ratio in the first decile compared to 47 percent for O/E ratio calculated by gnomAD (see Fig. 6a). If we instead of the LoF O/E ratio use the upper bound of the 90 confidence interval of the ratio (LOEUF) as suggested by Karczewski et al.[5] we see similar results (see Fig. 6b). We also compared Genovo and gnomAD using two lists of essential genes used in the gnomAD article[5] and obtained similar results (see Supplementary Fig. 4). If we look at the ability to classify whether a gene is haploinsufficient we get significantly higher AUC values using Genovo ($p$ value: 2.42 e-153, See Fig. 6c).

## Discussion

The k-mer pattern partition method introduced in this manuscript makes it possible to build robust models of the germline mutation process using k-mers. We have demonstrated the method by building models for each of the six-point mutation types as well as short insertions and deletions using de novo mutations as input. The results show that the out-of-sample error decreases as k increases proving that even nucleotides four bp away from a given position are informative about their mutation probability. An advantage of the pattern partition models is their interpretability, with overrepresented patterns readily available and not requiring any further analysis of the created models. The k-mer pattern partition approach is general and could, in principle, also be used to solve other problems where the input is small position-specific k-mers. The main shortcoming of the method in its current form is that the presented dynamic programming algorithm is exponential in running time which makes it infeasible to use for k-mers longer than nine bases. A possible improvement to alleviate this problem would be to consider heuristic algorithms that could quickly find a good (but not necessarily optimal) solution and thus enable analyses with larger k's. We do consider one such heuristic algorithm in the article (Supplementary Fig. 2), but it should be possible to come up with better heuristics that can balance accuracy with computational cost.

The predictive models created using kmerPaPa are not only robust, they have the added benefit that they are easily interpretable. The method can be seen as a dimensionality reduction method that reduces the high dimensional k-mer space to a lower

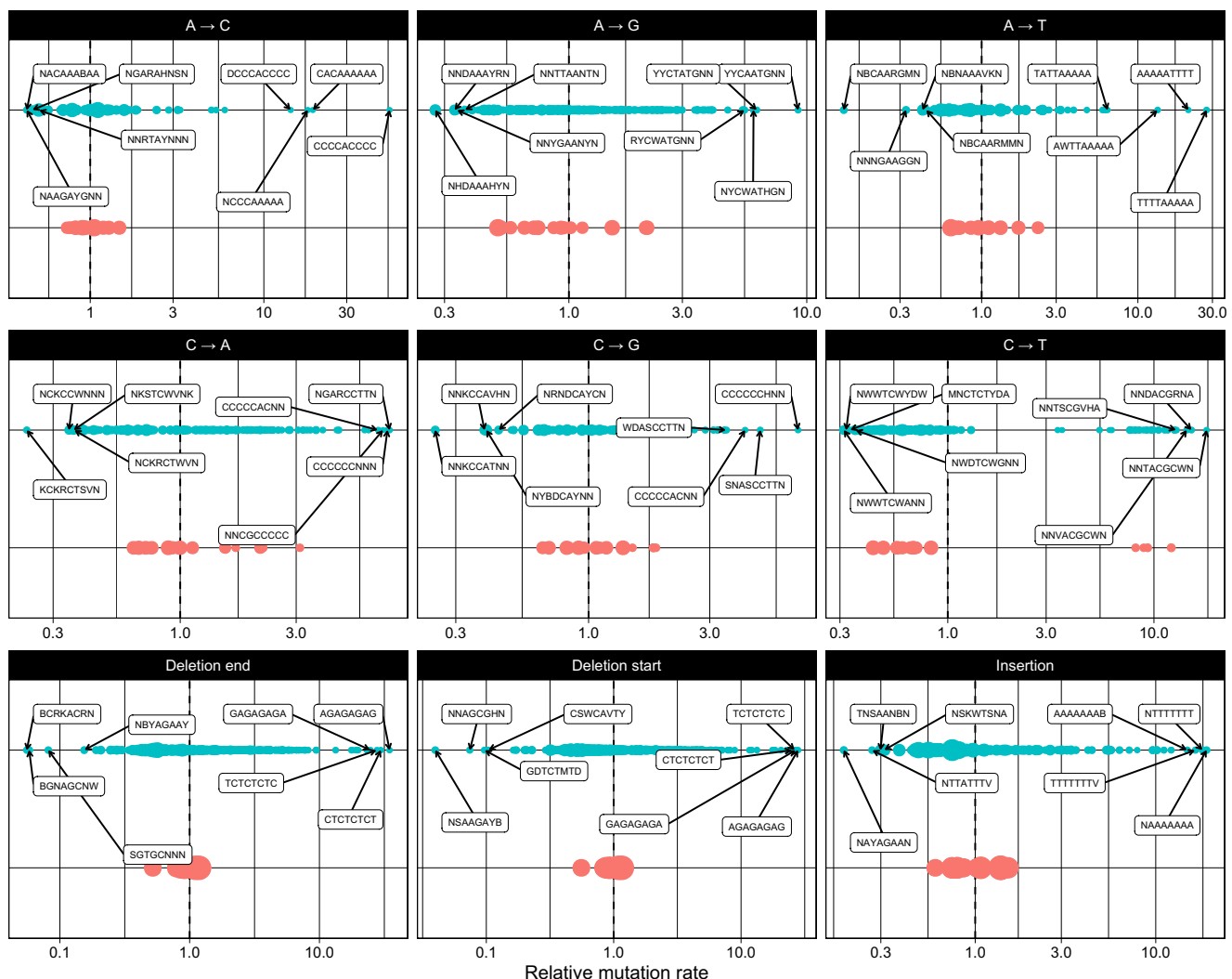

**Fig. 4 | The relative mutation rate for each of the patterns in the kmerPaPa models.** The blue points show the rate for the kmerPaPa patterns, the red points show the rate for each 3-mer. The dashed vertical line corresponds to the average mutation rate for the mutation type and the mutation rate values on the *x* axis are relative to this average and on a log scale. The size of the points reflects the number of genomic sites that match the pattern.

dimension space where each dimension is represented by an IUPAC pattern. This set of patterns with associated mutation rates makes it very easy to see the general differences between the high mutation rate k-mers and low mutation rate k-mers. Other methods have previously been used to create models of point mutation rates based on sequence context, but it is novel that we in this study also create robust models of indel mutation rates based on sequence context. To do that, we had to overcome the problem that the exact position of an indel often is indeterminable. We do this by enumerating all the possible positions and randomly choosing one of them for each indel. Future studies that train indel kmerPaPa models using larger input data sets and further examine the resulting patterns can hopefully increase our understanding of the indel mutation process.

Besides the k-mer pattern method implemented in the kmerPaPa software, we also present a tool called Genovo that can calculate the expected number of mutations with a specific functional consequence in a gene. Furthermore, Genovo can also calculate p-values for whether a gene contains significantly more observed mutations of a particular functional category than expected. This functionality can be used to find genes with causative de novo mutations in disorders such as autism and developmental disorders. We have compared Genovo to denovolyzeR, which contains similar functionality. Trying to predict the number of synonymous mutations in each gene reported in gnomAD we see that we do slightly better than denovolyzeR. Our ability to predict the number of frameshift mutations in a gene is, however, much better than denovolyzeR. This reflects that we train independent mutation rate models for insertions and deletions whereas denovolyzeR calculates the expected number of indel variants by assuming a correlation with the number of point mutations. Unlike denovolyzeR, which is based on a specific reference genome (hg19) and gene annotation, Genovo is more general and can calculate expectations and p-values for any reference genome and gene annotation. This allows users to use the newest reference genome and gene annotation at any time and makes it possible to use kmerPaPa and Genovo to analyze the enrichment or depletion of functional mutational categories in other species.

In conclusion, we have created a new robust method for predicting mutation probabilities based on sequence context for both point mutations and indels. Furthermore, we have developed a flexible software tool that makes it easy to use the created mutation rate models to find genes where de novo mutations cause disease or to measure the evolutionary constraint on genes.

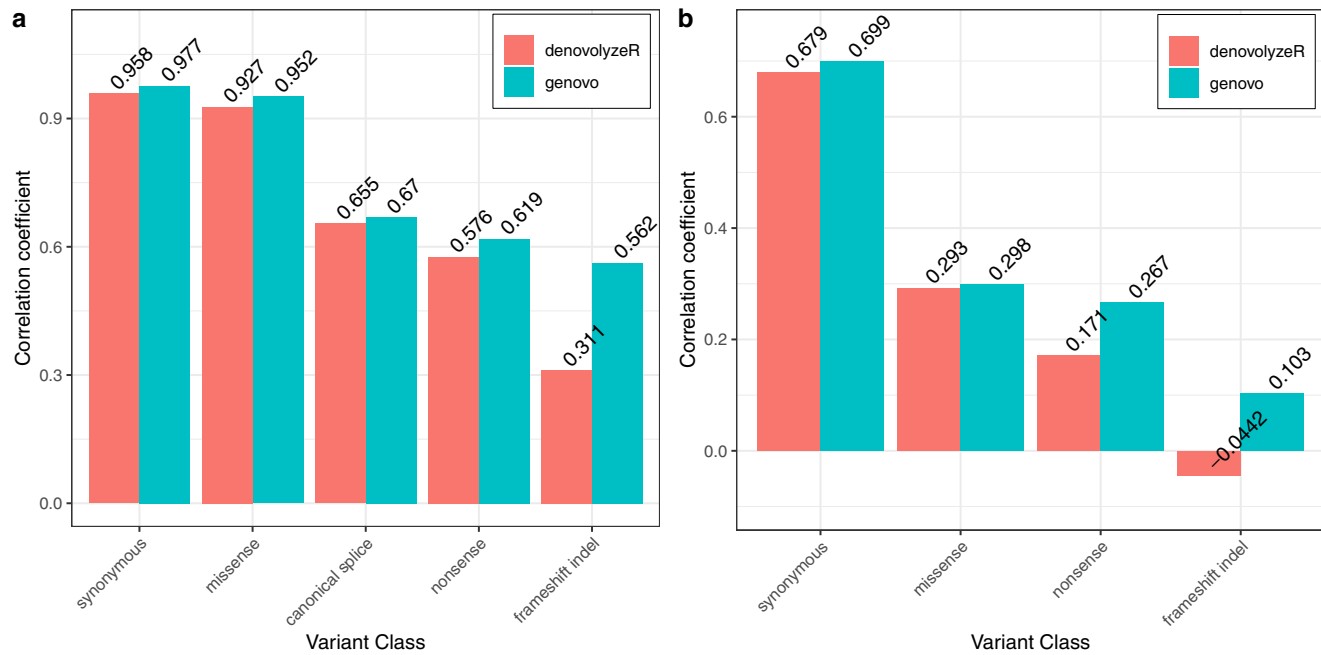

Fig. 5 | **Correlation between the predicted genic rate for a mutation type and the number of segregating variants of that type. a** Pearson correlation coefficient between the observed number of mutations per gene and the expected number using Genovo and denovolyzeR. **b** Pearson correlation coefficient between the observed number of mutations per coding position and the expected number per coding position for each gene using Genovo and denovolyzeR.

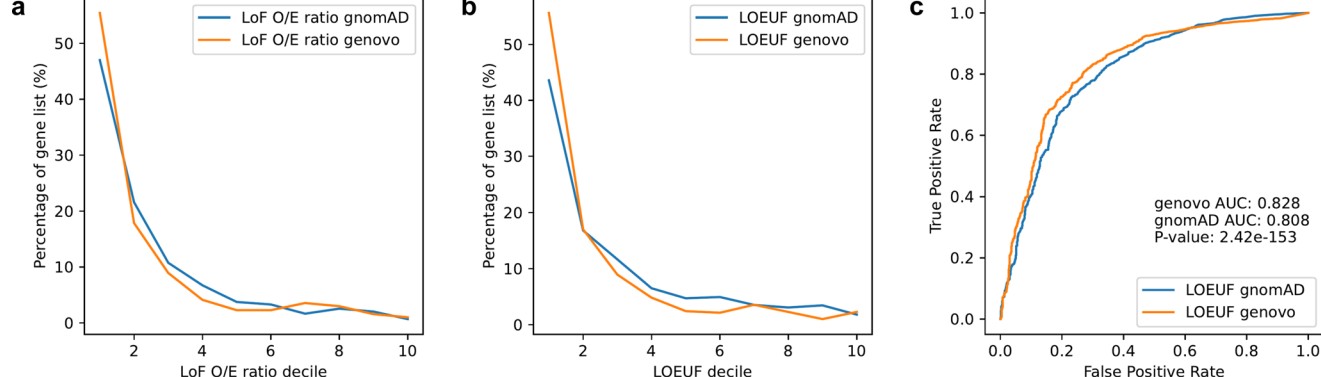

Fig. 6 | **Inferring constraint of haploinsufficient genes. a** Comparison of observed/expected ratio for loss of function variants (LoF O/E ratio) inferred by Genovo and gnomAD (blue and orange, respectively). The *x* axis depicts LoF O/E ratios sorted into deciles, whereas the *y* axis depicts the percentage of genes in each decile. Depicted numbers are shown in Supplementary Table 2. **b** Comparison of categorization by LOEUF scores inferred by Genovo - and LOEUF score by gnomAD.

The *x* axis depicts LOEUF scores sorted into deciles, whereas the *y* axis depicts the percentage of genes in each decile. Depicted numbers are shown in Supplementary Table 3. **c** Receiver Operating Characteristic (ROC) curve comparing the ability to predict whether a gene is annotated as haploinsufficient using LOEUF scores from gnomAD and Genovo.

## Methods

### Definition of k-mer pattern partition

A k-mer pattern partition is a set of IUPAC-encoded nucleotide patterns that partitions a set of k-mers so that each k-mer is matched by one and only one of the IUPAC patterns in the set. To find a pattern partition that groups k-mers with similar mutation rates as well as possible while not overfitting the data, we select the partition $P = \{p_1, ..., p_n\}$ that optimizes the following loss function:

$$\text{loss}(\{p_1, \ldots, p_n\}) = \sum_{i=1}^{n} (-2(M_i \log(r_i) + U_i \log(1 - r_i)) + c) \quad (1)$$

Where $M_i$ and $U_i$ are the number of mutated and unmutated positions that match the i'th pattern ($p_i$), and $c$ is a regularizing complexity penalty. And $r_i$ is the rate estimate for sites that match the i'th pattern regularized to be closer to the mean mutation rate, $\mu$, using a pseudo count, $\alpha$:

$$r_i = \left( \frac{M_i + \alpha}{M_i + U_i + \alpha/\mu} \right) \quad (2)$$

$M_i$, $U_i$, and $\mu$ can easily be calculated from the k-mer counts. If $m_i$ and $u_i$ denote the number of times k-mer $k_i$ has been observed mutated and unmutated, respectively, then we can calculate the

**Table 1 | The definition of all IUPAC characters and the list of all possible two-partitions for each IUPAC character**

| IUPAC code | Set | Two-partitions |
|---|---|---|
| S | {C,G} | (C,G) |
| W | {A,T} | (A,T) |
| R | {A,G} | (A,G) |
| Y | {C,T} | (C,T) |
| K | {G,T} | (G,T) |
| M | {A,C} | (A,C) |
| B | {C,G,T} | (C,G), (G,Y), (T,S) |
| D | {A,G,T} | (A,K), (G,W), (T,R) |
| H | {A,C,T} | (A,Y), (C,W), (T,M) |
| V | {A,C,G} | (A,S), (C,R), (G,M) |
| N | {A,C,G,T} | (A,B), (C,D), (G,H), (T,V), (R,Y), (S,W), (K,M) |

pattern counts by summing over the k-mers that match the pattern:

$$M_i = \sum_{k_j \text{ matches } p_i} m_j \qquad (3)$$

$$U_i = \sum_{k_j \text{ matches } p_i} u_j \qquad (4)$$

And $\mu$ is just the average rate:

$$\mu = \frac{\sum_i m_i}{\sum_i (m_i + u_i)} \qquad (5)$$

The loss function consists of the negative log-likelihood of the binomially distributed counts for each k-mer plus the regularization parameter $c$. This makes the loss function a generalization of several widely used model selection criteria. If we for instance use $c = 2$ the loss function would be the Akaike information criterion (AIC).

The two regularizing hyperparameters $c$, and $\alpha$ are fitted using cross-validation.

## Calculating the optimal pattern partition

Starting with the most general pattern, we can create any possible pattern partition by:

1. Picking a position in the pattern.
2. Splitting the IUPAC code at that position into two subcodes that form a partition (we call such a split a two-partition).
3. Possibly repeating these steps on each of the two sub-patterns.

Table 1 shows all the possible two-partitions for each IUPAC code.

Since all pattern partitions can be created using the strategy above, we can find the minimal loss for the pattern, $p$, using the following recursion formula:

$$f(p) = min\left(loss(\{p\}), \min_{i=1\ldots k}\left(\min_{x,y \in TwoPartition(p[i])}(f(p[:i] + x + p[i+1:]) + (f(p[:i] + y + p[i+1:])))\right)\right) \qquad (6)$$

Using this formula, we compute the optimal partitions bottom-up using dynamic programming so that the optimal partition for a given pattern is never calculated more than once. This algorithm's running time and memory usage are proportional to the number of possible patterns · and thus exponential in k. Our python implementation, speeded up using numba[18] for just-in-time compilation, can calculate the optimal 9-mer pattern partition for C→T mutations in ~4 h (given $\alpha$ and $c$). The size of the IUPAC alphabet is 15 (the four standard nucleotides plus the 11 characters shown in Table 1). This means that if

we wanted to calculate the optimal 11mer, we would need to multiply the 4 h running time with $15^2 = 225$, so that is not feasible using this algorithm.

Besides the optimal algorithm described above, we have also implemented a greedy heuristic algorithm where we only consider the most promising split of a given pattern into two and do not recursively test the other possible splits. This heuristic is much faster and consumes much less memory, which makes it possible to also test larger k-mers. Results shown in Supplementary Fig. 2 show that the improved speed comes with a cost of decreased performance. For point mutations, the models trained using the greedy algorithm were slightly worse, but for indel mutations, the models were much worse than those generated using the optimal algorithm. Only for the A→T mutation type there was a benefit of being able to use a higher k as the 11mer model did better than the 9-mer model for this mutation type. For the rest of the mutation types the results achieved with the optimal algorithm were clearly better than those achieved using the greedy algorithm. We thus decided to use the optimal algorithm to generate all models reported in this article.

## Estimating hyperparameters

To find the optimal $\alpha$ and $c$ we do cross-validation over a grid, and this means that we need to calculate the optimal pattern for each fold and each $\alpha$ and $c$ combination. This grid search is, however, easy to parallelize since each parameter combination can be run separately on different machines. To create the cross-validation folds the kmerpapa software uses a hypergeometric distribution to sample subsets of the k-mer count tables. For the models presented in this article, we used repeated 2-fold cross-validation where we created 5 independent cross-validation data sets for each parameter combination and selected the parameter combination that had the lowest average out-of-sample negative log-likelihood. Supplementary Fig. 5 compares the negative log-likelihood for different combinations of $\alpha$, $c$, and $k$. The results show that the optimal $c$ increases with $k$, which can explain why we for several of the mutation types end up with fewer parameters in the 9-mer model than the 7-mer model.

## De novo mutations and covered regions

To train kmerPaPa models, we use de novo mutations from 6 different WGS studies of parent-offspring trios[19–24]. Together these studies include 379330 autosomal point mutations from 7206 trios. Four of the six studies also report indel mutations (39,877 autosomal variants). Studies mapped to hg19 were first lifted to hg38.

Detection of de novo mutations requires strict filters, and usually there will be a substantial fraction of the genome where de novo mutations cannot be called due to insufficient coverage or low complexity sequence. We do not know the exact genomic regions where a mutation could have been called in each of the six studies. To deal with this, we have chosen a conservative set of regions—the strict callable mask from the 1000 genomes project[25]—and assume that mutations could have been called in this subset of the genome in all of the studies. Because we want to infer a genome-wide mutation model, we further exclude the C→G enriched regions described by[26] since they are known to have mutation patterns that differ substantially from the average. After removing the C→G enriched regions from the 1000 g callable, the callable regions contain 1754 million autosomal sites and after discarding all mutations that fall outside these regions, we are left with 249,437 autosomal SNVs and 22331 autosomal indels.

We use the program kmer_counter (https://github.com/BesenbacherLab/kmer_counter) to calculate the k-mer counts used as input kmerPaPa. As explained in the results section, we first test the kmerPaPa method by using the even-numbered chromosomes as training data and the odd-numbered chromosomes as test data. Afterwards, we train models using mutations on all chromosomes (see Supplementary Fig. 1). Besides training indel models using all

insertions and all deletions as presented in Fig. 4 we also train separate models for indels that are a multiple of 3 and indels that are not a multiple of 3. These models are used as input for the Genovo software to allow it to calculate separate probabilities for frameshift and in-frame indels.

## Computing pseudo $r^2$ values

To measure how well the models do on the test data we calculate Nagelkerke's pseudo $r^2$:

$$r^2_{\text{Nagelkerke}} = \frac{1 - e^{2(\ell_0 - \ell_M)/n}}{1 - e^{2\ell_0/n}} \qquad (7)$$

Where $l_M$ is the log-likelihood of a model where sites matching the $i$'th pattern is predicted to mutate with rate $r_i$:

$$\ell_M = \sum_{i=1}^{n} (M_i \log(r_i) + U_i \log(1 - r_i)) \qquad (8)$$

If $M_i$ and $U_i$ are the numbers of mutated and unmutated sites in the test data that match the $i$'th pattern. And $l_O$ is the log-likelihood of a model with no predictors.

The Nagelkerke's pseudo $r^2$ is equal to the Cox and Snell $r^2$ value divided by an adjustment that ensures that the maximum value is always 1. Because the pseudo $r^2$ values are a function of the number of data points in the data set, these statistics cannot be directly compared between type-specific models, where the number of data points varies. Due to the intrinsic randomness of the mutation processes, it will never be possible to predict exactly where mutations happen and get an $r^2$ value of 1.

## Genovo

From a list of observed mutations the Genovo software can determine the number of (1) synonymous, (2) missense, (3) nonsense, (4) start-codon disruption, (5) stop-codon disruption, (6) canonical splice site disruption (7) inframe indel, and (8) frameshift indel events in each transcript. Furthermore, the software takes the genomic sequence of every transcript and enumerates all possible point mutations. For each possible point mutation, it determines the type of point mutation and its probability derived from our kmerPaPa models. The possible mutations of each mutation type are tallied up to calculate the expected number of point mutations of each mutation type. In addition, the possible mutations and their probabilities are used to calculate a Poisson-binomial distribution of mutation events for every transcript, because each possible mutation event has a different probability to occur. We then sample from this distribution to calculate the p-value for the number of observed mutations of each mutation type and to calculate the confidence interval around the number of expected mutations (see Supplementary Fig. 3).

## Scaling mutation rates

The mutation rates output by k-merPaPa corresponds to the number of mutations expected in the number of individuals that we have data from. To turn these estimates into rates per generation we multiply all SNV rates with a factor so that the average mutation rate becomes $1.28 \times 10^{-8}$ per base per generation[26]. And we scale indel rates to have an average rate of $0.68 \times 10^{-9}$ per base per generation[27].

In Genovo these mutation rates should then be scaled to reflect the number of generations that the observed mutations correspond to. To avoid that some mutations get probabilities above one for data sets with many samples we use the following formula to scale the rate per base per generation, $r$, to get the probability that this mutation has

occurred in $n_{\text{gen}}$ observed generations:

$$r_{scaled} = 1 - (1 - r)^{2n_{gen}} \qquad (9)$$

For the gnomAD analysis where we do not have de novo mutations from a certain number of trios but segregating variants accumulated over an unknown number of generations, we set $n_{\text{gen}}$ so that the number of expected synonymous mutations fit the observed number of synonymous mutations in the whole exome.

## Correction for coverage

Coverage bias in exome sequencing data results in fewer sequenced individuals for some transcripts compared to others in the gnomADv2 data. This means that the expected number of mutations predicted by Genovo will be too high for genes that have low coverage. To correct for this we fit a lowess curve using the statsmodels package[28] in python, to a scatter plot with synonymous observed/expected values ($y$ axis) vs. the number of called samples ($x$ values) for each transcript (see Supplementary Fig. 6). The number of called samples for a transcript was estimated as the average AN (from the gnomAD2 vcf file) over all variants in the transcript. In the lowess-function, we used 0.05 as the fraction of the data used when estimating each $y$ value. Based on this inference, we made a cutoff where we only kept transcripts that have an average AN value above 50,000. For each transcript, we then divided the expected variant count with the corresponding lowess estimate for all variant classes to get a coverage corrected expectation.

## Running Genovo

To produce the expected variant count for each transcript we ran Genovo using hg19 and gencode v. 19 (same as used for gnomADv2) using a scaling factor of $2.2 \times 10^{-7}$. Then we applied the coverage correction as described above. For the comparison with denovolyzeR (version 0.2.0) shown in Fig. 5 we used the VEP annotations provided in "gnomad.exomes.r2.1.1.sites.vcf" to get the observed number of variants in each functional class.

Of the 4293 trios in the DDD data set 3664 contain at least one exome mutation. When splitting the data into test and training data sets we put half of the 3664 individuals with a mutation in each group. When calculating p-values using Genovo we did it using 10 million random samples from the null model. For both Genovo and denovolyzeR we do Bonferroni correction of the training set p-values based on the number of genes that have at least one observed LoF mutation in the training data (350 genes). For each gene, we only test the transcript used by denovolyzeR when running Genovo.

## Calculation of Genovo LOEUF score

When calculating the "loss-of-function observed/expected upper bound fraction" (LOEUF score)[5] from the output of the Genovo software we do it by dividing the number observed LoF variants by the lower bound of the 90% confidence interval of the expected number of LoF variants. Like the other expected values this lower bound used in the denominator had also been adjusted for coverage differences between transcripts as described above in the section *Correction for coverage*. As there were many transcripts with zero observed mutations, we added a pseudo score of 0.5 to all values of observed and expected LoF when we calculated the LoF O/E ratio.

## gnomAD and Genovo comparison

We used the pre-calculated LoF O/E ratios and LOEUF scores provided in the publicly available data provided by the gnomAD consortium (gnomad.v2.1.1.lof_metrics.by_transcript.txt.bgz) to compare O/E ratios and LOEUF scores between gnomAD and Genovo. We made sure that we compared transcripts that we had data for in both gnomAD and Genovo data sets. Thereafter, we split the transcripts into deciles and inferred how many transcripts were found in each decile. We did

this for each model−gnomAD or Genovo−and visually compared the two for three different groups of genes that were: essentiality of genes inferred by two lists (mouse gene knockout experiments and cellular inactivation assays) and haploinsufficient genes. We used the same curated gene lists that were used in used in the gnomAD paper[5] downloaded from https://github.com/macarthur-lab/gene_lists; haploinsufficient genes according to ClinGen Dosage Sensitivity Map (as of 13 Sep 2018)[17], genes deemed essential in mouse[29–31], genes deemed essential in human[32], and genes deemed essential and non-essential in human by CRISPR screening[33].

We conducted the ROC AUC analysis by using the sklearn.metrics.roc_curve function from the scikit-learn module[34] in python. For each of the AUC scores yielded for each prediction score (gnomAD or Genovo), we also conducted a DeLong test[35] to infer whether the AUC scores were significantly different. To conduct this test we used python code adapted from https://github.com/Netflix/vmaf/.

### Correlation between nonsense variants and frameshift variants
To calculate the correlations between the number of nonsense mutations and frameshift indels that we mention in the introduction we used two different data sets. For polymorphic variants we calculated the correlation based on gnomADv3. For de novo variants we used the de novo mutations reported in Halldorsson et al.[19]. In each case we calculated the Pearson correlation between the number of nonsense variants per coding site in each gene and the number of frameshift indels per coding site in each gene.

### Reporting summary
Further information on research design is available in the Nature Portfolio Reporting Summary linked to this article.

## Data availability
All de novo mutation data sets used as training data are publicly available. Two of them can be found as supplementary tables in the articles reporting them[19,20]. Two are available using links to external files provided in the articles[21,24]. The last two data sets[22,23] can be downloaded from denovo-db[36,37]. The gnomAD data can be downloaded from https://gnomad.broadinstitute.org. De novo mutations from the Deciphering Developmental Disorders consortium[16] were downloaded from supp table S1 in the article[16]. The strict mask from the 1000 genomes project can be downloaded here: http://ftp.1000genomes.ebi.ac.uk/vol1/ftp/data_collections/1000_genomes_project/working/20160622_genome_mask_GRCh38/StrictMask/20160622.allChr.mask.bed. The mutation rate models and predicted gene constraints generated in this study are provided as Supplementary Data. The kmerPaPa models trained on even-numbered chromosomes are available as Supplementary Data 1. The kmerPaPa models trained on all chromosomes are available as Supplementary Data 2. These models formatted as input files to Genovo can furthermore be found at https://github.com/BesenbacherLab/Genovo_Input. The predicted number of variants in each functional type for each gencode v19 transcript used for the gnomAD comparison are in Supplementary Data 3.

## Code availability
kmerPaPa can be installed from https://pypi.org/project/kmerpapa/ and the source code is available at https://github.com/BesenbacherLab/kmerPaPa. Genovo can be installed from https://crates.io/crates/genovo and the source code is available at https://github.com/BesenbacherLab/genovo. kmer_counter can be installed from https://pypi.org/project/kmer-counter/ and the source code is available at https://github.com/BesenbacherLab/kmer_counter. The adapted python code used to calculate the DeLong test comparing ROC-AUCs is available at https://github.com/BesenbacherLab/ROC-utils.

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

## Acknowledgements

This work was funded by a Sapere Aude–Starting Grant (no. 4181-00490) (to S.B.) from the Independent Research Fund Denmark.

## Author contributions

S.B. designed and implemented the kmerPaPa software. J.B. designed and implemented the Genovo software. S.B. ran the kmerPaPa software to produce mutation rate models and the Genovo software to produce predicted rates per mutation type per transcript. A.K. performed coverage correction and the gnomAD comparison analysis. S.B. and A.K. wrote the manuscript. All authors reviewed and approved the final manuscript.

## Competing interests

The authors declare no competing interests.
