## [Peer Review File · Nature Communications]

A method to build extended sequence context models of point mutations and indelsREVIEWER COMMENTS

Reviewer #1 (Remarks to the Author):

Bethune and colleagues introduce a novel method for modeling the effects of sequence context on mutation rates of point mutations and indels. This model uses a clever recursive algorithm to partition the $N=4^{(k-1)}$ possible k-mers into $\ll N$ informative subtypes by grouping k-mers according to similar patterns that allow for ambiguity at each position, denoted using standard IUPAC codes. This model also allows for modeling context-specific mutation rates of indels, which have typically been ignored in models of context-dependent mutation rates. Finally, the authors introduce a separate tool, Genovo, that uses this model to predict the expected number of functional mutation types in each gene.

This will absolutely be of interest to researchers in the field, as it directly addresses some well-known limitations of modeling the effects of sequence context on mutability. Consideration for indels is a nice bonus too, and the manuscript is nicely situated in the recent literature on using detailed mutation models to more accurately quantify the functional constraint of genes. The paper is well-written and the methods are well-conceived and thoroughly described. I do not have any major concerns with the paper, just a few suggestions to flesh out the analyses a bit.

Fig. 2a: it's interesting that the number of parameters in the kmerPaPa model actually decreases when going from 7-mers to 9-mers—essentially, I take this to mean that 1) most of the time, bases +/-4bp from the focal site provide little information about mutability of the focal site (e.g., if they're not informative at all, the number of parameters in the 9-mer partition model should be identical to that of the 7-mer partition model) but 2) in some cases the identity of the bases +/-4bp from the focal site may be *more* informative than that of multiple bases closer to the focal site.

For example, consider a possible subset of 4 subtypes we might observe in a 7-mer based partitioning model:

SAA[C>T]GGG

WAA[C>T]GGG

ASA[C>T]GGG

AWA[C>T]GGG

These could be partitioned into a maximum of 64 possible 9-mers, but as few as just 4 9-mers, if the base in the +/-4bp position is entirely uninformative.

To achieve *fewer* parameters in the 9-mer model than the 7-mer model, however, we would need to observe a partitioning pattern where the identity of a base +/-4bp is more informative than at multiple other bases closer to the focal site, e.g. suppose the 9-mer partitioning model yields the following 2 subtypes that subsume all 4 of the 7-mer subtypes above:

ANNN[C>T]GGGN

BNNN[C>T]GGGN

I'd love to see a deeper characterization of these sorts of cases—do they occur non-randomly with respect to the bases at the +/-4bp position, the basic mutation type, or the identity of other flanking bases? This could potentially yield some interesting biological insights that might have been overlooked in the saturated K-mer models used by Carlson et al. and Aggarwala & Voight (e.g., perhaps this is mainly a feature of CpG>TpG sites, and the -4bp position is highly predictive of methylation status at the CpG?) More generally, it would be interesting to find a way to summarize the differences between the saturated models and the kmerPaPa models—maybe using some sort of sequence logo plots showing which bases in the k-mer tend to get summarized by IUPAC codes?

Methods: the authors indicate that extending the kmerPaPa model beyond 9-mers is not computationally feasible. I'd like to see some more consideration for how the model could actually be modified to allow for longer k-mers. Given that increasing the sequence context is typically expected to produce diminishing returns, it seems like a hierarchical model might be a tractable alternative. E.g., given the 71 subtypes found in the 3-mer model, re-run for the 71*225 possible 5-mer subtypes (considering only the +/-2 position), and so on. This limits the model by essentially enforcing a constraint that prevents the above scenario (e.g., the number of parameters in a [k+2]-mer model will always be greater than or equal to the number of parameters in a k-mer model), so we'd expect its predictive performance suffers slightly. However, this should run in linear(ish) time complexity rather than exponential, potentially enabling kmerPaPa to be tested head-to-head with a recent deep-learning mutation model trained on 21-mer motifs (<https://www.biorxiv.org/content/10.1101/2021.10.25.465689v1>).

(One could also consider any number of variations that balance predictive performance with computational cost, such as introducing the hierarchical strategy starting with the results of the current 9-mer model, fixing the 400 9-mer subtypes and evaluating the 400*225 possible 11-mer subtypes—this is just a suggestion, but if it's easy to do, might make for a nice supplementary figure or two!).

Minor comments:

Introduction: “While some correlation exists between the number of polymorphic nonsense mutations and the number of polymorphic frame-shift indels, this correlation is primarily due to selection and not mutation”—include a citation for this

Fig. 4: labels that overlap the vertical dashed line are hard to read (try putting `geom_vline()` before `geom_label_repel()` in the `ggplot` statement)

Methods: I’m assuming computation time should increase by 15^2 -fold when going from 9-mers to 11-mers—is it because there are 4 bases + 11 IUPAC codes = 15 possibilities to consider at the ± 5 bp positions? Make this more explicit.

The Github repository for `kmerPaPa` currently states that it has been merged into the `GeNovo` repository—the Code Availability statement should be updated accordingly. Also, perhaps consider revising the manuscript throughout to refer to the k-mer pattern partition model and the functional prediction tool as modules of the `GeNovo` software package—maybe something like “`GeNovo-KPP`” and “`GeNovo-FP?`” The paper feels a bit disjointed to describe these as two distinct tools.

Reviewer #2 (Remarks to the Author):

This review is focused on the computational side due to my background.

The authors propose a k-mer based technique to estimate germline mutation rate based on sequence context.

A strength of the method is the ability to incorporate indels and also slightly longer k-mers than before. Previous methods focused on SNPs and shorter k-mers (up to 7). Though, the method is limited to k values around 9, as acknowledged by the authors. So the increase in k-mer length is moderate (only 2 bp) and the novelty appears to reside in the support for indels. This seems valuable for frameshifts. An additional contribution of the study is the `Genovo` software which performs statistics on the expected number of mutations. It is a nice application of the method and demonstrates its usability for finding genes with *de novo* mutations involved in disorders. Results also match predictions made by `gnomAD`, apart from the first and 9th deciles (Sup Fig 3), interestingly.

I could test the `kmerPaPa` code and it ran well, good job there. For `Genovo`, I could install but not test (see remarks below). Overall, this appears to be an interesting set of contributions around germline mutation process models.

Major remarks:

1. More details are needed in the Methods section "Definition of k-mer pattern partition":
 - 1a. Please explain how M_i, U_i and "mu" can be determined. (E.g. referencing later sections.)
 - 1b. Please explain how the cross-validation procedure works to determine "alpha" and "c" e.g. using pseudocode.
 - 1c. Please explain the rationale for defining the loss function as it is.
2. In the kmerPaPa github repo, please explain the steps allowing one could re-create the input .txt files.
3. Genovo github repo: please provide example data for users to be able to execute a test run.
4. Please briefly explain the Nagelkerke r^2 coefficient for the non-statistician readership. Are higher values better? Is 0.06 considered "reasonable"? Some more background on this would be welcome.

Minor remarks:

- * Fig 3a: is it an unfortunate coincidence that the deletion start k-mers are the same as the deletion end k-mers?
- * Please rephrase to avoid an awkward ending of this sentence : "The ability to predict the number of expected mutations in a gene is not only relevant for finding genes where a specific cohort has an enriched number of functional mutations."
- * In section "Scaling mutation rates", some numbers are oddly formatted, e.g. " $1.28 \times 10^{-8} 26$ ", perhaps due to a citation.
- * Please provide software versions, e.g. for denovolyzeR.
- * Regarding abstract sentence: "Revealing that for some mutation types, the mutation rate of a position is significantly affected by nucleotides that are up to four base pairs away." The wording of the beginning of the sentence is awkward, and also, wasn't this fact known by previous methods already? since they considered k up to 5.
- * Could you please comment why the 1st and 9th deciles of Sup Fig 3 of Genovo do not match gnomAD?

Reviewer #1 (Remarks to the Author):

Bethune and colleagues introduce a novel method for modeling the effects of sequence context on mutation rates of point mutations and indels. This model uses a clever recursive algorithm to partition the $N=4^{(k-1)}$ possible k-mers into $\ll N$ informative subtypes by grouping k-mers according to similar patterns that allow for ambiguity at each position, denoted using standard IUPAC codes. This model also allows for modeling context-specific mutation rates of indels, which have typically been ignored in models of context-dependent mutation rates. Finally, the authors introduce a separate tool, Genovo, that uses this model to predict the expected number of functional mutation types in each gene.

This will absolutely be of interest to researchers in the field, as it directly addresses some well-known limitations of modeling the effects of sequence context on mutability. Consideration for indels is a nice bonus too, and the manuscript is nicely situated in the recent literature on using detailed mutation models to more accurately quantify the functional constraint of genes. The paper is well-written and the methods are well-conceived and thoroughly described. I do not have any major concerns with the paper, just a few suggestions to flesh out the analyses a bit.

Response:

Thanks for the nice comments and thorough review.

Fig. 2a: it's interesting that the number of parameters in the kmerPaPa model actually decreases when going from 7-mers to 9-mers—essentially, I take this to mean that 1) most of the time, bases +/-4bp from the focal site provide little information about mutability of the focal site (e.g., if they're not informative at all, the number of parameters in the 9-mer partition model should be identical to that of the 7-mer partition model) but 2) in some cases the identity of the bases +/-4bp from the focal site may be *more* informative than that of multiple bases closer to the focal site.

For example, consider a possible subset of 4 subtypes we might observe in a 7-mer based partitioning model:

```
SAA[C>T]GGG  
WAA[C>T]GGG  
ASA[C>T]GGG  
AWA[C>T]GGG
```

These could be partitioned into a maximum of 64 possible 9-mers, but as few as just 4 9-mers, if the base in the +/-4bp position is entirely uninformative.

To achieve *fewer* parameters in the 9-mer model than the 7-mer model, however, we would need to observe a partitioning pattern where the identity of a base +/-4bp is more informative than at multiple other bases closer to the focal site, e.g. suppose the 9-mer partitioning model yields the following 2 subtypes that subsume all 4 of the 7-mer subtypes above:

```
ANNN[C>T]GGGN  
BNNN[C>T]GGGN
```

I'd love to see a deeper characterization of these sorts of cases—do they occur non-randomly with respect to the bases at the +/-4bp position, the basic mutation type, or the identity of other flanking bases? This could potentially yield some interesting biological insights that might have been overlooked in the saturated K-mer models used by Carlson et al. and Aggarwala & Voight (e.g., perhaps this is mainly a feature of CpG>TpG sites, and the -4bp position is highly predictive of methylation status at the CpG?) More generally, it would be interesting to find a way to summarize the differences between the saturated models and the kmerPaPa models—maybe using some sort of sequence logo plots showing which bases in the k-mer tend to get summarized by IUPAC codes?

Response:

We have looked into the differences between the 7mer models and the 9mer models. And we do not see any obvious examples where it looks like the +/-4 bp is more informative than bases closer to the focal point. Instead

the main reason for the smaller number of parameters in the 9mer models compared to the 7mer models is that we select the hyperparameters independently using Cross Validation and we see that the optimal value of the hyperparameter c increases with k . If we use the same c values for the 7mer model and the 9mer models the 9mer model do get tend to have more parameters. The reason why the Cross Validation leads to higher values of c for the 9mer models compared to the 7mer models must be that the parameter space is much larger, which increases the risk of overfitting. We now include a new figure (Supp. Fig. S5) showing the results of the grid search and mention that c increases with k in the new subsection called “Estimating hyperparameters”.

Methods: the authors indicate that extending the kmerPaPa model beyond 9-mers is not computationally feasible. I'd like to see some more consideration for how the model could actually be modified to allow for longer k-mers. Given that increasing the sequence context is typically expected to produce diminishing returns, it seems like a hierarchical model might be a tractable alternative. E.g., given the 71 subtypes found in the 3-mer model, re-run for the 71×225 possible 5-mer subtypes (considering only the +/-2 position), and so on. This limits the model by essentially enforcing a constraint that prevents the above scenario (e.g., the number of parameters in a $[k+2]$ -mer model will always be greater than or equal to the number of parameters in a k -mer model), so we'd expect its predictive performance suffers slightly. However, this should run in linear(ish) time complexity rather than exponential, potentially enabling kmerPaPa to be tested head-to-head with a recent deep-learning mutation model trained on 21-mer motifs (<https://www.biorxiv.org/content/10.1101/2021.10.25.465689v1>).

(One could also consider any number of variations that balance predictive performance with computational cost, such as introducing the hierarchical strategy starting with the results of the current 9-mer model, fixing the 400 9-mer subtypes and evaluating the 400×225 possible 11-mer subtypes—this is just a suggestion, but if it's easy to do, might make for a nice supplementary figure or two!).

Response:

To test a strategy that balance predictive performance with computational cost, we have now implemented a greedy heuristic algorithm that only consider the most promising split of a given pattern into two and do not recursively test the other possible splits. The results comparing this algorithm to our optimal algorithm is shown in Supp. Figure S2. We appreciate the suggestion of the hierarchical model and think that it would probably be superior to the greedy heuristic we have now tested - but the optimal algorithm would still be superior to both when it comes to predictive ability (but not running time). Given a set of hyperparameters it is possible to run kmerPaPa on each of the sub-patterns of an existing pattern partitioning but automating the hierarchical strategy and integrating it with the cross validation is more cumbersome and we have decided to leave that for future development.

Minor comments:

Introduction: “While some correlation exists between the number of polymorphic nonsense mutations and the number of polymorphic frame-shift indels, this correlation is primarily due to selection and not mutation”—include a citation for this

Response:

We were not able to find a good citation for this. Instead we now provide evidence for the claim ourselves by comparing the correlation between nonsense mutations and frame shift indels among segregating variants and de novo variants. The paragraph now reads as follows:

“While some correlation exists between the number of polymorphic nonsense mutations and the number of polymorphic frame-shift indels ($r=0.493$, see methods) there is no observable correlation between the number of de novo nonsense mutations and de novo frame-shift indels ($r=-0.003$, see methods). This lack of correlation among de novo variants indicates that the correlation between nonsense mutations and frame-shift indels in segregating variants is primarily due to selection and not mutation”

Fig. 4: labels that overlap the vertical dashed line are hard to read (try putting `geom_vline()` before `geom_label_repel()` in the ggplot statement)

Response:

Thanks for noticing this. We have followed your suggestion and it does improve the readability.

Methods: I'm assuming computation time should increase by 15^2 -fold when going from 9-mers to 11-mers—is it because there are 4 bases + 11 IUPAC codes = 15 possibilities to consider at the +/-5bp positions? Make this more explicit.

Response:

Yes, that is correct. We have tried to clarify this in the methods section.

The Github repository for kmerPaPa currently states that it has been merged into the GeNovo repository—the Code Availability statement should be updated accordingly. Also, perhaps consider revising the manuscript throughout to refer to the k-mer pattern partition model and the functional prediction tool as modules of the GeNovo software package—maybe something like “GeNovo-KPP” and “GeNovo-FP?” The paper feels a bit disjointed to describe these as two distinct tools.

Response:

By mistake, the first version of the genovo software was put in a repository named kmerpapa (<https://github.com/jbethune/kmerPaPa>), but it is not (and has never been) the same repository as the kmerpapa repository (<https://github.com/BesenbacherLab/kmerPaPa>). The genovo and kmerpapa software are written in different languages (rust and python, respectively), and it is possible to use the kmerpapa software for tasks that are not related to the genovo software. For these reasons, we prefer to keep the two software packages in separate repositories with different names.

Reviewer #2 (Remarks to the Author):

This review is focused on the computational side due to my background.

The authors propose a k-mer based technique to estimate germline mutation rate based on sequence context. A strength of the method is the ability to incorporate indels and also slightly longer k-mers than before. Previous methods focused on SNPs and shorter k-mers (up to 7). Though, the method is limited to k values around 9, as acknowledged by the authors. So the increase in k-mer length is moderate (only 2 bp) and the novelty appears to reside in the support for indels. This seems valuable for frameshifts. An additional contribution of the study is the Genovo software which performs statistics on the expected number of mutations. It is a nice application of the method and demonstrates its usability for finding genes with de novo mutations involved in disorders. Results also match predictions made by gnomAD, apart from the first and 9th deciles (Sup Fig 3), interestingly.

I could test the kmerPaPa code and it ran well, good job there. For Genovo, I could install but not test (see remarks below). Overall, this appears to be an interesting set of contributions around germline mutation process models.

Response:

Thank you for bringing the missing test data for genovo to our attention. We have now added a test data set (see response to point 3 below).

Major remarks:

1. More details are needed in the Methods section "Definition of k-mer pattern partition":

1a. Please explain how M_i, U_i and “mu” can be determined. (E.g. referencing later sections.)

Response:

We have now added formulas describing how to calculate M_i, U_i and mu to the methods section.

1b. Please explain how the cross-validation procedure works to determine “alpha” and “c” e.g. using pseudocode.

Response: We have now added a subsection to the methods section called “Estimating hyperparameters” that explains how we create the cross-validation folds and select the best parameter combination. We have also added a supplementary figure showing the results of the CV grid search.

1c. Please explain the rationale for defining the loss function as it is.

Response:

We have added the following clarification to help readers understand the loss function:

“This loss function consists of the negative log-likelihood of the binomially distributed counts for each k-mer plus the regularization parameter c . This makes the loss function a generalization of several widely used model selection criteria. If we for instance use $c=2$ the loss function would be the Akaike information criterion (AIC).”

2. In the kmerPaPa github repo, please explain the steps allowing one could re-create the input .txt files.

Response:

We have now added a subsection called “Creating input data” to the kmerPaPa README file that explains how to create input files.

3. Genovo github repo: please provide example data for users to be able to execute a test run.

Response:

We have now included an example data set in the github repo and have added a subsection called “Running the whole pipeline on a test data set” to the README file.

4. Please briefly explain the Nagelkerke r^2 coefficient for the non-statistician readership. Are higher values better? Is 0.06 considered “reasonable”? Some more background on this would be welcome.

Response:

We have now added a sub-section titled “Computing pseudo r^2 values” to the methods section where we explain how the Nagelkerke r^2 are calculated and discuss the limitations and expectations of such values.

Minor remarks:

* Fig 3a: is it an unfortunate coincidence that the deletion start k-mers are the same as the deletion end k-mers?

Response:

No. The start k-mers are the reverse complement of the end k-mers, so it is to be expected that the results are very similar. We could have chosen to use only the start or the end k-mers but we don't think the results get worse by considering both and then taking the average of the two predictions when we make predictions.

* Please rephrase to avoid an awkward ending of this sentence : “The ability to predict the number of expected mutations in a gene is not only relevant for finding genes where a specific cohort has an enriched number of functional mutations.”

Response:

We have rephrased the two first sentences of this paragraph to improve the readability.

* In section “Scaling mutation rates”, some numbers are oddly formatted, e.g. “ $1.28 \times 10^{-8} 26$ ”, perhaps due to a citation.

Response: The problem should be fixed now.

* Please provide software versions, e.g. for denovolyzeR.

Response: We have now done so.

* Regarding abstract sentence: "Revealing that for some mutation types, the mutation rate of a position is significantly affected by nucleotides that are up to four base pairs away." The wording of the beginning of the sentence is awkward, and also, wasn't this fact known by previous methods already? since they considered k up to 5.

Response: We have now changed the beginning of the sentence. Since it is the middle position of a k-mer that is mutated a 5mer only considers positions that are up to two base pairs away. Whereas a 9mer considers positions four bases away.

* Could you please comment why the 1st and 9th deciles of Sup Fig 3 of Genovo do not match gnomAD?

Response:

We assume you refer to the decile plots in Supp, fig. S4. The fact that the 1st decile of the essential and knockout gene list is larger for genovo than the gnomAD metrics is a good sign. That means that genovo is better at ranking the genes so that the most essential/constrained get the lowest values. With regards to the 9th decile we assume you are thinking about why the 9th decile is higher than the 10th decile for the non-essential genes. That is more puzzling and unfortunately we do not know the reason behind this.

REVIEWERS' COMMENTS

Reviewer #1 (Remarks to the Author):

The authors have thoroughly addressed all of my previous comments in this revision.

Reviewer #2 (Remarks to the Author):

The authors provided very satisfactory corrections. I was able to test the software. I only have two minor remarks. I trust that the authors will address them satisfactorily and I do not need to see another revision.

Regarding my earlier question: ("Fig 3a: is it an unfortunate coincidence that the deletion start k-mers are the same as the deletion end k-mers? (Response: "No. The start k-mers are the reverse complement of the end k-mers, so it is to be expected that the results are very similar. We could have chosen to use only the start or the end k-mers but we don't think the results get worse by considering both and then taking the average of the two predictions when we make predictions")")

I am now confused by this response because the start k-mers (TCGC, GCGC) are not the reverse complement of the end k-mers (TCGC, GCGC), only GCGC is. I now believe that there is an error in the Figure and the first deletion end k-mer should read GCGA instead of TCGC.

Minor detail: please make the test data unpacks in a separate folder, and remove the .* files.

Response to reviewer comments.

Reviewer #1 (Remarks to the Author):

The authors have thoroughly addressed all of my previous comments in this revision.

Reviewer #2 (Remarks to the Author):

The authors provided very satisfactory corrections. I was able to test the software. I only have two minor remarks. I trust that the authors will address them satisfactorily and I do not need to see another revision.

Regarding my earlier question: ("Fig 3a: is it an unfortunate coincidence that the deletion start k-mers are the same as the deletion end k-mers? (Response: "No. The start k-mers are the reverse complement of the end k-mers, so it is to be expected that the results are very similar. We could have chosen to use only the start or the end k-mers but we don't think the results get worse by considering both and then taking the average of the two predictions when we make predictions")")

I am now confused by this response because the start k-mers (TCGC, GCGC) are not the reverse complement of the end k-mers (TCGC, GCGC), only GCGC is. I now believe that there is an error in the Figure and the first deletion end k-mer should read GCGA instead of TCGC.

Response:

Even though we expect the patterns at the start of deletions to look like the patterns at the end of deletions, we do not expect them to be identical. We count the start points and end points separately and thus also sample one of the possible indel positions independently for deletion_start and deletion_end k-mers. So there is no error in the figure - the patterns are similar but not identical. We now show the 4 patterns with highest and lowest rate instead of only 3 in Figure 4. This makes it easier to see the similarities between then deletion_start and deletion_end patterns.

Minor detail: please make the test data unpacks in a separate folder, and remove the .* files.

Response:

Thanks for the advice. We will do so.